# Improving the Neural GPU Architecture for Algorithm Learning

## Abstract

Algorithm learning is a core problem in artificial intelligence with significant implications on automation level that can be achieved by machines. Recently deep learning methods are emerging for synthesizing an algorithm from its input-output examples, the most successful being the Neural GPU, capable of learning multiplication. We present several improvements to the Neural GPU that substantially reduces training time and improves generalization. We introduce a new technique - hard nonlinearities with saturation costs - that has general applicability. We also introduce a technique of diagonal gates that can be applied to active-memory models. The proposed architecture is the first capable of learning decimal multiplication end-to-end.

## 1. Introduction

Several architectures have appeared that are capable of learning algorithms of moderate complexity. Neural GPU (Kaiser & Sutskever, 2015) is the most promising among the proposed architectures for algorithm learning because it is the only one capable of learning multiplication that generalizing to inputs much longer than the training examples. However, it is fragile since only a tiny fraction of the trained models generalize well.

In this paper, we study ways to improve the Neural GPU to obtain faster training and better generalization. The proposed improvements allow us to achieve substantial gains: the model can learn binary multiplication in 800 steps versus 30000 steps that are needed for the original Neural GPU, and, most importantly all the trained models generalize to 100 times longer inputs with less than 1% error. The model can also learn a wider range of problems with similar generalization performance, e.g. the decimal multiplication, which is the first time it has been learned end-to-end. To learn decimal multiplication we use a different representation where each decimal digit is encoded in binary.

The improvements that achieve these goals are introduction of nonlinearities with saturation cost and introduction of a diagonal gating mechanism. We also improve the training

schedule by training on all input lengths simultaneously and use a larger learning rate with AdaMax optimizer (Kingma & Ba, 2014). We integrate gradient clipping into AdaMax.

We analyze the impact of each improvement separately and show that all of them are relevant to the achieved performance. We find that using hard nonlinearities with saturation cost is the key factor to achieve good generalization.

## 2. Related Work

Recurrent networks are the simplest devices capable of computation on arbitrary length inputs. It is employed in LSTM(Hochreiter & Schmidhuber, 1997) and GRU (Cho et al., 2014) networks for sequence classification. Thy scale with sequence length but each cell has a constant amount of memory that essentially limits the learnable problems to regular languages. Grid LSTM (Kalchbrenner et al., 2015) allow explicit unrolling along time and memory dimension and are able to learn more complex tasks such as addition and memorization.

Simple algorithms such as sequence copying and reversal can be learned with the current reinforcement learning techniques (Zaremba & Sutskever, 2015; Zaremba et al., 2016). (Graves et al., 2014) developed a Neural Turing Machine capable of learning and executing simple programs such as repeat copying, simple priority sorting, and associative recall. (Graves et al., 2016) improve it for solving more complex tasks.

(Joulin & Mikolov, 2015) introduce differentiable stack and double linked list data structures. Pointer Networks (Vinyals et al., 2015) use soft attention and generalize to a variable-sized output space depending on the input sequence length. This model was shown to be effective for combinatorial optimization problems such as the traveling salesman and Delaunay triangulation. Neural Random-Access Machines (Kurach et al., 2016) introduce a memory addressing scheme potentially allowing constant time access due to discretization. (Grefenstette et al., 2015) introduce more neural data structures and evaluate them on several sequence processing tasks. A hierarchical memory layout with logarithmic access time is introduced in (Andrychowicz & Kurach, 2016) with both differentiable and reinforcement learning versions being presented. A different setting of algorithm learning

is explored in (Reed & De Freitas, 2015) where a model is trained on execution traces instead of input and output pairs; this richer supervision allows to induce higher level programs. See also (Kant, 2018) for a recent overview of the existing approaches.

Neural GPU (Kaiser & Sutskever, 2015; Kaiser & Bengio, 2016) seems the most promising approach since it is simple and fast and can learn fairly complicated algorithms such as addition and binary multiplication. But only a small fraction of the trained models generalize to instances of unbounded length. The authors train 729 models to find one that generalizes well. (Price et al., 2016) is able to train Neural GPU on decimal multiplication by using curriculum learning when the same model is trained at first for binary multiplication then for base-4 and only then for decimal.

## 3. The Model

Neural GPU(NGPU) was introduced by (Kaiser & Sutskever, 2015). It is a recurrent network with a multi-dimensional state where a Convolution Gated Recurrent Unit (CGRU) is applied to the state at every time-step. CGRU is a combination of convolution operation and GRU(Cho et al., 2014) which computes the state $s_t$ at time $t$ from the state at time $t-1$ according to the following rules:

$$s_t = u_t \odot s_{t-1} + (1 - u_t) \odot c_t$$
$$c_t = \tanh(U * (r_t \odot s_{t-1}) + B)$$
$$u_t = \sigma(U' * s_{t-1} + B')$$
$$r_t = \sigma(U'' * s_{t-1} + B'')$$

In the above equations, $U$, $U'$, $U''$ are convolution kernel banks, $B$, $B'$, $B''$ are bias vectors; these are the parameters that will be learned. $U * s$ denotes a convolution of a kernel bank $U$ with a state $s$; $u \odot s$ denotes element-wise vector multiplication and $\sigma$ is the sigmoid function.

Given an input of length $n$, it is embedded into the first state, each symbol independently, producing a state with its first dimension equal to $n$, then CGRU is applied to it several times, and output is read from the last state by using a softmax for each symbol.

We use a 2-dimensional state of shape [$n$,$m$] where $n$ is the length of input and $m$ is the number of maps. The convolution kernel banks are of shape [3,$m$,$m$] and we fix the filter length to 3. We confirmed experimentally that this is the optimal setting for all considered tasks. We use $n$ applications of the convolutional unit, all with the same set of parameters.

The gating mechanism incorporated in the CGRU facilitates data copying to the same cell in the next time-step. This is essential for bringing together features separated in time during training. However, for most tasks, it is also required to bring together features from both ends of the input. There-

fore, we introduce gates that copy data to a neighboring cell in the next time-step. We call these diagonal gates. We split all maps of a state into 3 parts $s_t = (s_t^1, s_t^2, s_t^3)$. The first part has a gate from the same cell in the previous time-step as in a CGRU, the second part uses gate from the left neighbor cell, and the third part uses gate from the right neighbor cell. To implement the diagonal gates, we need to shift the parts $s_{t-1}^2$ and $s_{t-1}^3$ to the right and left respectively and then apply a CGRU to the result. Shifting can be conveniently expressed as a convolution. Right shift corresponds to convolution with filter [1,0,0], left shift to convolution with [0,0,1] and no shift to convolution with [0,1,0]. We define the Diagonal Convolutional Gated Recurrent Unit (DCGRU), which we will use instead of CGRU, as follows:

$$s_t = u_t \odot \tilde{s}_t + (1 - u_t) \odot c_t$$
$$\tilde{s}_t = (\tilde{s}_t^1, \tilde{s}_t^2, \tilde{s}_t^3)$$
$$\tilde{s}_t^1 = s_{t-1}^1 * [0, 1, 0]$$
$$\tilde{s}_t^2 = s_{t-1}^2 * [1, 0, 0]$$
$$\tilde{s}_t^3 = s_{t-1}^3 * [0, 0, 1]$$

Definitions of $u_t$ and $c_t$ are the same as for CGRU. The division of maps into 3 parts is only conceptual; an implementation uses a depthwise convolution operating directly on $s_{t-1}$ convolving each map with the required convolution filter independently.

Using gates that operate in different directions is not a novel idea. A similar mechanism is used in Grid LSTM (Kalchbrenner et al., 2015) where different units perform gating along different dimensions of the grid. But introduction such gates in a convolutional architecture is new.

(Kaiser & Sutskever, 2015) have found that introducing gate cutoff improves performance. We go further and use hard_tanh and hard_sigmoid functions for all nonlinearities in the DCGRU. They are piecewise linear approximations of tanh and $\sigma$, namely

$$\text{hard\_tanh}(x) = \max(-1, \min(1, x))$$
$$\text{hard\_}\sigma(x) = \max(0, \min(1, (x+1)/2))[1]$$

To keep the units away from saturation we add an extra cost to the loss function. For one unit we define

$$\text{saturation\_cost}(x) = \min(0, |x| - s\_limit)$$

with a parameter $s\_limit$ slightly less than 1 to keep the unit in its linear range. A value $s\_limit = 0.9$ works well in our case. We calculate the saturation cost for each application of hard_tanh($x$) and hard_$\sigma$($x$), sum all of them together and add to the loss function with an appropriately small weight. We choose the weight such that the total saturation cost is 100 times smaller than the error loss.

---

[1]Other literature may contain a slightly different definition, but the following one is preferable in our case since we can use the same saturation cost for both functions.

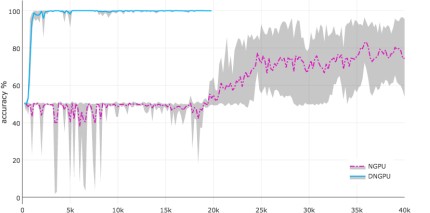

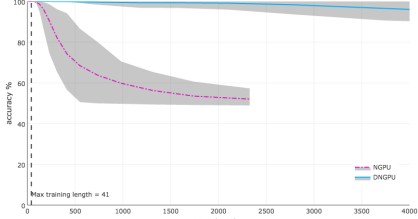

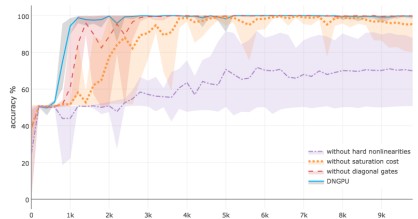

*Figure 1.* Accuracy on test set length 401 vs. step on binary multiplication.

*Figure 2.* Accuracy on inputs of different lengths. The vertical dashed line shows the training length.

*Figure 3.* Ablation study of the proposed features.

We train the model on inputs of all lengths simultaneously. As in the original architecture, we instantiate several bins of different lengths and place each training example into the smallest bin it fits and pad the remaining length. However, instead of training each bin separately, we sum their losses together and use one optimizer for the total loss. In this way, we avoid scheduling of bins and obtain faster convergence since, typically, several bins contribute to progress at each training step. We do not use parameter sharing relaxation of NGPU since we find in unnecessary.

We use initial learning rate $lr = 0.005$ and decrease it if no progress is made for 600 steps. We use AdaMax optimizer (Kingma & Ba, 2014); it has a strong guarantee that each parameter value will change by no more than $lr$ at each step. We integrate gradient clipping into AdaMax optimizer. We clip each variable separately to the range proportional to its decayed maximum that is used internally by the optimizer. In this way, we do not need to set some predetermined clipping threshold. We use gradient noise of magnitude proportional to the learning rate as suggested in (Neelakantan et al., 2015). We apply dropout only to the update vector $c_t$ of the CGRU as proposed in (Semeniuta et al., 2016). Such dropout helps to avoid memory loss over many time steps.

## 4. Evaluation

We have implemented the proposed architecture in tensorflow. The code is available on GitHub[2]. In this section, we compare the proposed architecture (denoted by DNGPU) with the original architecture (denoted NGPU) by (Kaiser & Sutskever, 2015) and evaluate individual improvements of DNGPU proposed in this paper. We choose multiplication task as the basis for the evaluation since it is the most complex of the tasks considered in (Kaiser & Sutskever, 2015). On other tasks including addition and sorting our architecture performs the same or better.

For comparison we use NGPU implementation provided by

---

[2]To appear in final version.

the authors (Kaiser & Sutskever, 2015). We set the number of maps $nmaps = 24$, dropout probability = 0.09, other parameters leaving at their default values.

For DNGPU we use the number of maps $m = 96$ to match the data amount carried in one state of NGPU (which use 24 maps in 4 rows). We use learning rate $lr = 0.05$, and dropout probability = 0.1.

We use essentially the same training set as in the (Kaiser & Sutskever, 2015) consisting of 10000 examples of every length up to 41 (two 20 bit numbers are multiplied). We trained 5 models with a random initialization and measured their accuracy on a test set containing random inputs of length 401 (two 200 bit numbers are multiplied). In this way, we can show both training speed and generalization in one graph. We used a computer with Intel Xeon E312 2.4 GHz processor, 64GB RAM and a Tesla K40 GPU card for testing.

### 4.1. Performance and generalization

To compare DNGPU with NGPU, we plot the accuracy of both models on the test set for each step of training, see Fig 1. The solid lines show the average of all runs and the shaded area shows the scatter among different runs. Accuracy is defined as the percentage of correctly predicted output bits over all examples. We can see that the DNGPU converges much faster and achieves near 100% accuracy in all runs. It requires only about 800 steps to reach 99% accuracy.

To explore generalization beyond length 401, see Fig 2 which shows the accuracy of both architectures depending on input length. We see that DNGPU generalizes much better. All trained DNGPU models exceeded 90% accuracy, and two out of five exceeded 99% accuracy on length 4001.

To summarize, our architecture outperforms the original by a wide margin both in terms of training speed and of generalization. Our implementation consistently reaches 99% accuracy on the test set in less than 15 minutes. The original NGPU trains slower and achieves 90% accuracy only on some runs. Our findings about NGPU are consistent

with a much more massive evaluation in (Neelakantan et al., 2015) Table 6 which shows that only a small fraction of its trained instances generalize well to length 401. Note that generalization of both models can be improved by increasing dropout probability together with the number of maps.

## 4.2. Ablation study

We tried to understand how much each of the proposed enhancements contributes to the improved performance. Fig 3 shows how the model performs when one of the proposed features is turned off. The magenta line shows the effect of using traditional soft tanh and sigmoid instead of hard ones. The yellow line shows performance with hard nonlinearities but without saturation cost. The red line shows performance without diagonal gates. The blue line is the suggested architecture which performs the best. A model trained without hard nonlinearities leads to especially poor performance. A closer look reveals that these models managed to fit the training set but generalized poorly to long test instances. The same figure shows that hard nonlinearities without saturation cost also perform poorly. Training is unstable and does not converge to 100% accuracy.

## 4.3. Decimal multiplication

Our model can learn base-4 multiplication with consistently good generalization if we increase the number of maps to 192. However, like our predecessors, we did not succeed on the decimal multiplication task in its originally proposed form. But our architecture can learn decimal multiplication if we encode each decimal digit in binary. We use 4 bits per digit and mark the start of each digit with a different encoding of its first bit. Such encoding produces 4 times longer inputs and outputs. We implemented this encoding in input/output data generation part, but equivalently it can be implemented inside the Neural GPU itself by appropriate adjustment of its input and output layers.

For evaluation, we increased the number of maps $m$ to 192 and performed training on examples of length 41 (multiplication of two 5 digit decimal numbers) and tested on examples of length 401 (multiplication of two 50 digit decimal numbers) as before. Fig 4 shows the results.

We were surprised to see that it generalized so well despite training only on very short examples containing two 5 digit numbers. Additionally, two models out of 5 generalized to length 401 with less than 1% error. Of course, for better generalization we have to train on longer inputs. The binary encoding allows easier training. However, it comes with a significant overhead, i.e., a 4x increase in the input length which leads to a 16x increase in the unrolled model and a proportional increase training time and memory requirements.

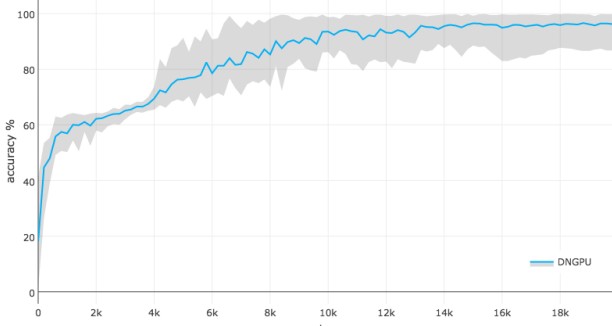

*Figure 4.* Accuracy on test set length 401 vs. step on decimal multiplication.

## 5. Conclusions

We have presented several improvements to the Neural GPU architecture that substantially decrease training time and improve generalization. The main improvements are hard nonlinearities with saturation cost and a diagonal gating mechanism. We have shown that the hard nonlinearities with saturation cost contribute the most to obtaining better generalization. They may find further applications also in ordinary reccurent networks such as LSTM and GRU.

A larger learning rate together with AdaMax optimizer also helps the training performance, but the introduced saturation cost is essential to keep the learning convergent.

The improved architecture can easily learn a variety of tasks including the binary multiplication on which other architectures struggle. If we increase the number of maps to 192, we can also learn base-4 multiplication with consistently good generalization. Furthermore, if we encode the decimal input/output digits in binary, the architecture can also learn decimal multiplication end-to-end.

The improved architecture is considerably simpler than the original NeuralGPU, enabling an easier extension to handle harder problems. One such possible extension could be scaling the model to solve tasks requiring more than $n$ slots of memory or more than $n$ time steps. Simply enlarging the size of the model did not work well. So we leave the question of proper scaling of the model for future work.

The correct generalization of the learned models to arbitrary large inputs is still an open problem, and it is not even clear why some models generalize, and others do not. With the proposed simpler model and faster training, it will be possible to address this question more effectively.

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

## Appendix 1

To see that diagonal gates have impact, we can inspect execution trace of a trained model on some input. An execution trace is a collection of state values arising in the computation over all time steps. It is visualized as an image for each map where the input is given at the top, and the result is read from the bottom row of the image. In the figure below we can see all 96 maps of an execution trace performing binary multiplication on two 50 digit random numbers. We can notice computing patterns that are aligned with the gate direction. Every 4 image rows correspond to maps with a different gate direction.

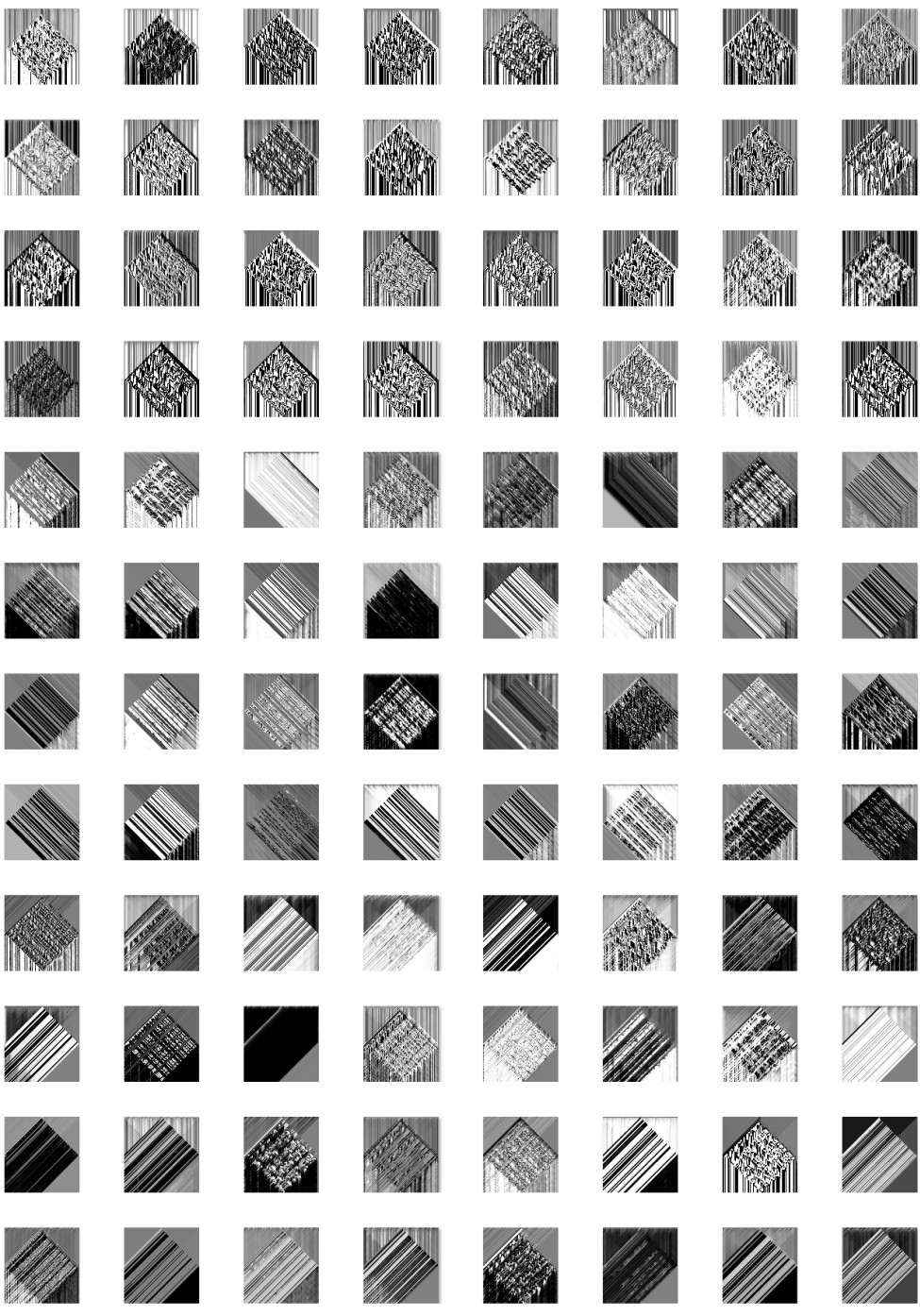

## Appendix 2

All 48 maps of an execution trace performing sorting where 100 numbers in range 0 to 5 are sorted. We can notice computing patterns that are aligned with the gate direction. Every 16 images correspond to maps with a different gate direction.

# Appendix 3

We have tested the proposed architecture on all tasks given in (Kaiser & Sutskever, 2015). Multiplication was the hardest task which was explored in detail in this paper. The second hardest was sorting. Here we give a chart showing the accuracy of our model on the sorting task where natural numbers in range 0 to 5 are being sorted.

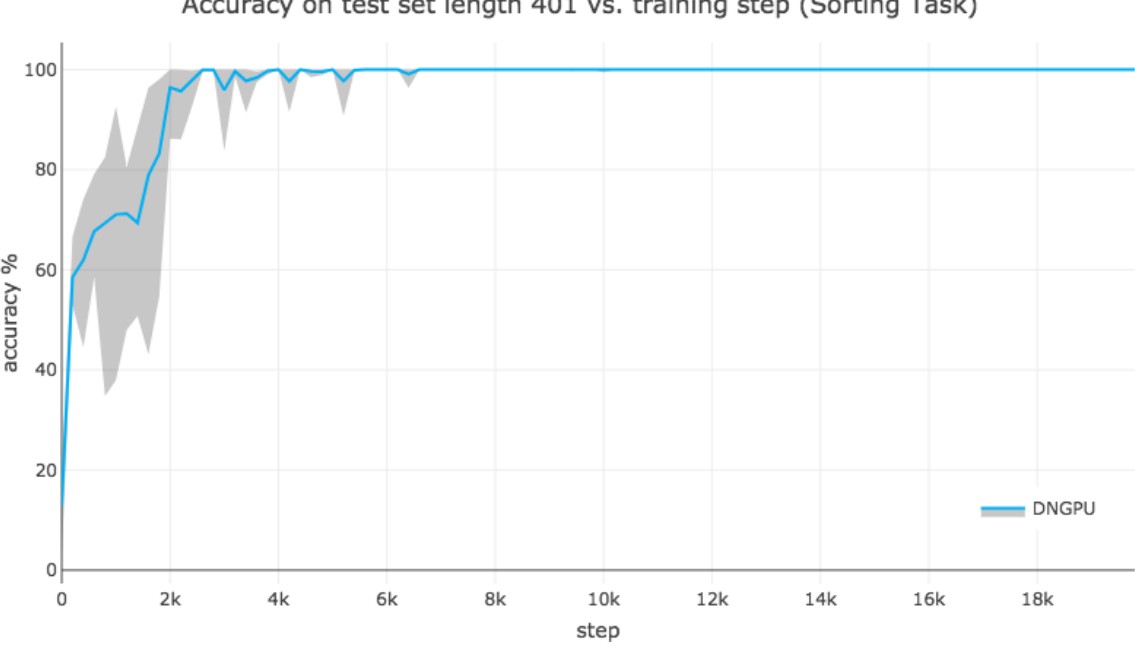