# OpenReview forum: "Improving the Neural GPU Architecture for Algorithm Learning"
_ICML.cc/2018/Workshop/NAMPI — NAMPI 2018_

### Review · AnonReviewer1 · 2018-06-17
**Hard tanh and hard sigmoid improve generalization**

**Rating:** 9
**Confidence:** 4

**Review:**

The paper modifies Neural GPU to use hard tanh and hard sigmoid instead of tanh and sigmoid.
This modification together with a saturation cost leads to generalization to much longer sequences.
E.g., the network was able to generalize to sequences of length 4001 even if training only on sequences with lengths up to 41.

I recommend the paper for acceptance.
The ablation experiments are informative, the description is clear and the improved Neural GPU can be a starting point for future improvements.

Questions:
1) The tanh in GRU is preventing to copy the state unchanged, if not using the hard tanh. Have you tried LSTM instead of GRU?
2) When clipping the gradient, what was the used "range proportional to its decayed maximum"?
3) For the decimal multiplication, would it be enough to make the input 4-times longer and the depth 4-times deeper?
The net may be able to learn the mapping from decimal to binary in the input and output layers.

Comments on evaluation:
4)  It may look unfair to use only nmaps=24 in the NGPU and nmaps=96 in the DNGPU.
I assume that you ablations experiment with no enhancements is equivalent to NGPU with nmaps=96.

Suggestions to improve clarity:
5) When describing GRU or the diagonal gates, it would be more readable to list the program lines in the execution order.
E.g., start with r_t, u_t and end with c_t, s_t.
6) When describing the gradient clipping, I guess, you are missing "gradient" in the following sentence:
"We clip [the gradient of] each variable separately ..."
6) When encoding the decimal digits to 4 bits, the following sentence is unclear: "We use 4 bits per digit and mark the start of each digit with a different encoding for it first bit."
I'm happy that the details will be visible in the released code.

---

> ### Comment · ~Karlis_Freivalds1 · 2018-07-05
> **Answers**
>
> * You are right - we can learn decimal multiplication if we add 3 blank symbols after each decimal digit and increase the network depth 4 times. Binary encoding of digits is not necessary.
>
> * The reference NGPU implementation uses two-dimensional state with its last dimension fixed to 4. We use one-dimensional state. So the information carried in NGPU state is 4*24=96 which matches our implementation. Therefore comparison is fair.

---

### Review · AnonReviewer3 · 2018-06-25
**A great step above the Neural GPU**

**Rating:** 9
**Confidence:** 5

**Review:**

The authors present DNGPU which greatly improves upon the original Neural GPU. From the results, it is currently the best architecture for algorithm learning so the paper should absolutely be accepted. Several ablation studies could be added, e.g., is AdaMax essential for making it work? But it's a clear accept, great paper!

---

### Review · AnonReviewer2 · 2018-06-25
**Improving Neural GPU Architectures**

**Rating:** 8
**Confidence:** 5

**Review:**

Summary: This paper presents an extension over the Neural GPU architecture that helps to eliminate some of the tricks required to train Neural GPUs and achieve better results. They introduce a hard-saturating activation function that makes the training of the model much easier and stable. In order to avoid the activations of the model get stuck in the saturation regime they introduce a smart trick that penalizes the activations if it is very close to margin.

Pros:
1) The extensions makes sense. The hard-saturating activation functions make the training of the NGPU model much easier.
2) The results are quite good and convincing.

Cons:
1) Writing requires some more work. In particular, model description part of the paper is not clear enough.

Questions:
* Have you tried this with LSTMs instead of GRUs?
* For NGPU baseline did you use all the tricks mentioned in the original paper, i.e. dropout, weight-sharing relaxation?
* How crucial is the Adamax for the training? Have you tried training NGPU baseline with Adamax as well?
* Have you tried batch-norm as well?

---

> ### Comment · ~Karlis_Freivalds1 · 2018-07-04
> **Answers**
>
> * LSTM can also be used successfully. The proposed hard nonlinearities and diagonal gates work very well with it. Using LSTM in place of GRU in our experiments yields similar results. We prefer GRU because it is simpler.
>
> * For NGPU baseline we use all the tricks which are found in its published source code.
>
> * The proposed model can be successfully trained also with Adam optimizer, only a smaller learning rate about 0.001 should be used.
>
> * Application of batch normalization is not straightforward since layer shape depend on input length. A modification where normalization is averaged over the length dimension shows improved training(less number of steps to reach certain accuracy) but generalization to longer inputs fails. We tried also LayerNorm on the nmaps dimension. It works and correctly generalizes but gives marginal improvement.

---

### Decision · ~NAMPI_Admin1 · 2018-06-28
**Paper1 Final Decision**

Accept